# Graphene on SiC Substrate as Biosensor: Theoretical Background, Preparation, and Characterization

**DOI:** 10.3390/ma14030590

**Published:** 2021-01-27

**Authors:** Alexander A. Lebedev, Sergey Yu Davydov, Ilya A. Eliseyev, Alexander D. Roenkov, Oleg Avdeev, Sergey P. Lebedev, Yurii Makarov, Mikhail Puzyk, Sergey Klotchenko, Alexander S. Usikov

**Affiliations:** 1Solid State Electronic Department, Ioffe Institute, 194021 St. Petersburg, Russia; sergei_davydov@mail.ru (S.Y.D.); lebedev.sergey@mail.ioffe.ru (S.P.L.); 2Solid State Physics Department, Ioffe Institute, 194021 St. Petersburg, Russia; Ilya.Eliseyev@mail.ioffe.ru; 3Nitride Crystals Group, 194156 St. Petersburg, Russia; roenkov47@yandex.ru (A.D.R.); oleg_avdeev65@mail.ru (O.A.); yuri.makarov@nitride-crystals.com (Y.M.); 4Faculty of Lazer Photonics and Optoelectronics, ITMO University, 197101 St. Petersburg, Russia; alexander.usikov@nitride-crystals.com; 5Chemical Faculty, Herzen University, 191186 St. Petersburg, Russia; puzyk@mail.ru; 6Molecular Biology of Viruses Department, Smorodintsev Research Institute of Influenza, 197376 St. Petersburg, Russia; osfatik@mail.ru

**Keywords:** graphene, SiC, sublimation, Auger and Raman spectroscopies, Green-function method, graphene gas sensor, grapheme biosensor

## Abstract

This work is devoted to the development and optimization of the parameters of graphene-based sensors. The graphene films used in the present study were grown on semi-insulating 6H-SiC substrates by thermal decomposition of SiC at the temperature of ~1700 °C. The results of measurements by Auger and Raman spectroscopies confirmed the presence of single-layer graphene on the silicon carbide surface. Model approach to the theory of adsorption on epitaxial graphene is presented. It is demonstrated that the Green-function method in conjunction with the simple substrate models permit one to obtain analytical results for the charge transfer between adsorbed molecules and substrate. The sensor structure was formed on the graphene film by laser. Initially, a simpler gas sensor was made. The sensors developed in this study demonstrated sensitivity to the NO_2_ concentration at the level of 1–0.01 ppb. The results obtained in the course of development and the results of testing of the graphene-based sensor for detection of protein molecules are also presented. The biosensor was fabricated by the technology previously developed for the gas sensor. The working capacity of the biosensor was tested with an immunochemical system constituted by fluorescein and monoclonal antibodies (mAbs) binding this dye.

## 1. Introduction

The discovery of graphene and the study of its properties is one of the brightest pages in the development of solid-state physics over the past 15 years. This discovery sparked the emergence of a whole class of two-dimensional structures, of which graphene is still the most-studied material. The exfoliation method, used by K. Novoselov and A. Geim in their first work on the preparation and study of graphene, is reduced to the separation of a one-atom-thick flake from a graphite crystal. Until now, graphene samples obtained by this technology have had the best structural perfection. It was on these samples that the most important results were obtained, which confirmed the two-dimensional nature of this material. However, their small size and irregular and unpredictable geometrical shape in advance do not allow the exfoliation method in industry.

Graphene films obtained by thermal destruction of the surface of silicon carbide are the second in terms of structural perfection. Thus, it is possible to obtain structures up to industrially important dimensions; in this case, the dimensions are limited only by the initial SiC substrate, i.e., up to 6 inches (150 mm) in diameter.

Theoretical and experimental studies show that graphene has a unique set of electrophysical properties:High mobility of charge carriers in combination with their low concentration;The maximum possible ratio of surface area to volume;Low noise level.

The combination of these properties leads to the fact that the addition of a minimal amount of impurity to the graphene surface can noticeably change the conductivity of the graphene film. Thus, graphene is a very promising material for the manufacture of various types of sensors.

In this work, we discuss the technology of forming a biosensor based on a graphene film obtained by thermal destruction of the surface of silicon carbide. A brief review of the model approach to the theory of adsorption on epitaxial graphene is presented. New results obtained by detecting various types of biological molecules with a graphene sensor are analyzed.

## 2. Theoretical Background: Models of Adsorption on Epitaxial Graphene

After appearing in a previous paper [1], where it was shown that using graphene as a substrate makes it possible to detect a single gas molecule, a great interest in the performance of graphene-based gas sensors [2,3,4] and biosensors [4,5,6,7,8,9] arose. To fulfill this program, corresponding adsorption theory is needed (see, for example, [10,11,12,13] and references on the earlier papers therein). Most of the theoretical works are first-principal numerical calculations, based on the different versions of (DFT). Here we will describe model approaches to the problem using Green’s function method [14].

We begin with the adsorption on single-layer free-standing graphene. As far as we know, the first corresponding model was put forward in [15,16]. In this model, graphene’s density of state (DOS) is given as $\rho_{G}\left(\omega \right)=c\left|\omega \right|/t^{2}$ for |ω|≤t, c/|ω| for t<|ω|≤3t and 0 for |ω|>t, where ω is the energy variable, t is the nearest-neighbor hopping energy, c=2/(1+2ln3) is the normalized factor, and the zero energy corresponds to the Dirac point. It was proposed that the adsorbed atom or molecule can be considered as one-electron (one-hole) adparticle, characterized by single orbital $\varepsilon_{a}$, whose coupling to substrate is $V_{a}$. This model gives adparticle’s DOS as the Lotentz-type function of the form
(1)$$\rho_{a}\left(\omega \right)=\frac{1}{\pi }\frac{\Gamma_{a}\left(\omega \right)}{{\left(\omega -\varepsilon_{a}-\Lambda_{a}\left(\omega \right)\right)}^{2}+\Gamma_{a}^{2}\left(\omega \right)}$$
where quasilevel’s width and shift functions are $\Gamma_{a}\left(\omega \right)={\pi V}_{a}^{2}\rho_{G}\left(\omega \right)$ and $\Lambda_{a}\left(\omega \right)=\pi^{-1}P\int_{-\infty }^{\infty }\Gamma_{a}\left(\omega^{\prime }\right){\left(\omega -\omega^{\prime }\right)}^{-1}d\omega^{\prime }$, where P means principal integral value (the exact expression for $\Lambda_{a}\left(\omega \right)$ is given in [15,16]).

At zero temperature the occupation number $n_{a}$ of adparticle’s quasilevel is given by the sum of the band contribution $n_{b}=\int_{-3t}^{E_{F}}\rho_{a}\left(\omega \right)d\omega$ and local state contribution $n_{l}={\left|1-\partial \Lambda \left(\omega \right)/\partial \omega \right|}_{\omega_{l}}^{-1}$, where $E_{F}$ is the Fermi level and the energy of local state $\omega_{l}$ is the root of the equation $\omega -\varepsilon_{a}-\Lambda \left(\omega \right)=0$ for ω<−3t [15]. If initially (before adsorption) level $\varepsilon_{a}$ was occupied, then the adparticle charge is $Z_{a}=1-n_{a}$; if initially this level was empty, then $Z_{a}=-n_{a}$. In the first case, an electron hops from adparticle onto graphene (donor), and in the second case, a hole hops from adparticle onto graphene (acceptor). Value of charge transfer is equal to $\left|Z_{a}\right|$.

Now turn to the epitaxial graphene (epigraphene) model description. The simplest approach is described in [17]. The first step here is the low-energy approximation for the free-standing graphene electron dispersion: ε(q)=±(3at/2)|q|, where a = 1.42 Å is the nearest-neighbor distance, q is the wave-vector separation from the Dirac point vector $K=\left(2\pi /3\sqrt{3}a\right)\left(1,\sqrt{3}\right)$ [18]. Such a dispersion gives the density of states ${\rho^{\prime }}_{G}\left(\omega \right)=2\left|\omega \right|/\xi^{2}$ for |ω|≤ξ and 0 for |ω|>ξ, where cut-off energy $\xi =\sqrt{2\pi \sqrt{3}}t$. It is easy to see that ${\rho^{\prime }}_{G}\left(\omega \right)$ is the simplified version of the $\rho_{G}\left(\omega \right)$ from [15,16].

General expression for the epigraphene DOS is given by
(2)$${\tilde{\rho }}_{G}\left(\omega \right)=\frac{\Gamma_{G}\left(\omega \right)}{{\pi \xi }^{2}}\left[\ln\frac{{\left(\Omega \mp \xi \right)}^{2}+\Gamma_{G}^{2}\left(\omega \right)}{\Omega^{2}+\Gamma_{G}^{2}\left(\omega \right)}+\frac{2\Omega }{\Gamma_{G}\left(\omega \right)}\left(\arctan\frac{\Omega }{\Gamma_{G}\left(\omega \right)}-\arctan\frac{\Omega \mp \xi }{\Gamma_{G}\left(\omega \right)}\right)\right]$$

Here $\Omega =\omega -E_{0}-\Lambda_{G}\left(\omega \right)$, $\Lambda_{G}\left(\omega \right)=\pi^{-1}P\int_{-\infty }^{\infty }\Gamma_{G}\left(\omega^{\prime }\right){\left(\omega -\omega^{\prime }\right)}^{-1}d\omega^{\prime }$, $\Gamma_{G}\left(\omega \right)={\pi V}_{G}^{2}\rho_{\mathrm{sub}}\left(\omega \right)$, where $V_{G}$ is the energy of the graphene-substrate interaction and $\rho_{\mathrm{sub}}\left(\omega \right)$ is the substrate DOS. In what follows, we will consider SiC as a substrate and use the Haldane–Anderson model $\rho_{\mathrm{sub}}\left(\omega \right)=\bar{\rho }$ for $\left|\omega -E_{0}\right|\geq E_{g}/2$ and 0 for $\left|\omega -E_{0}\right|<E_{g}/2$, where $E_{0}$ is the center of energy gap $E_{g}$ position relative to the Dirac point. This DOS corresponds to the shift function $\Lambda_{G}\left(\omega \right)=\left({\bar{\Gamma }}_{G}/\pi \right)\ln\left|\left(\omega -E_{0}-E_{g}/2\right)/\left(\omega -E_{0}+E_{g}/2\right)\right|$, where ${\bar{\Gamma }}_{G}={\pi V}_{G}^{2}\bar{\rho }$. Estimations of $E_{0}$ values for the different SiC polytypes are given in [19].

There are two limiting regimes for the graphene–substrate interaction: Strong coupling, when $t/\Gamma_{G}<<1$, and weak coupling, when $t/\Gamma_{G}>>1$. In the first case, ${\tilde{\rho }}_{G}\left(\omega \right)$ tends to the DOS of adsorbed single carbon adatom, while in the second case ${\tilde{\rho }}_{G}\left(\omega \right)\to \rho_{G}\left(\omega \right)$ (quasi-free-standing graphene). More rigorous expressions for ${\tilde{\rho }}_{G}\left(\omega \right)$ in both regimes are given in [17]. It is clear that only the second case is of practical interest. Thus, below we will consider only weak coupling regime.

It is easy to understand now that for the DOS of particle adsorbed on epigraphene, formula (1) has to be rewritten in the form
(3)$${\tilde{\rho }}_{a}\left(\omega \right)=\frac{1}{\pi }\frac{{\tilde{\Gamma }}_{a}\left(\omega \right)}{{\left(\omega -\varepsilon_{a}-{\tilde{\Lambda }}_{a}\left(\omega \right)\right)}^{2}+{\tilde{\Gamma }}_{a}^{2}\left(\omega \right)}$$
where ${\tilde{\Gamma }}_{a}\left(\omega \right)={\pi \tilde{V}}_{a}^{2}{\tilde{\rho }}_{G}\left(\omega \right)$, ${\tilde{\Lambda }}_{a}\left(\omega \right)=\pi^{-1}P\int_{-\infty }^{\infty }{\tilde{\Gamma }}_{a}\left(\omega^{\prime }\right){\left(\omega -\omega^{\prime }\right)}^{-1}d\omega^{\prime }$, and ${\tilde{V}}_{a}$ is the adparticle–epigraphene interaction. Then the occupation number of adparticle’s quasilevel is ${\tilde{n}}_{a}=\int_{-\infty }^{E_{F}}{\tilde{\rho }}_{a}\left(\omega \right)d\omega$. It was shown in [20] that the electronic state of the adparticle and charge transfer are affected by both the graphene and substrate. The prevailing effect should be determined for each particular adsorption system.

Up to now, we considered adsorption of a single particle at zero temperature. For finite coverages $\Theta =N_{a}/N_{\mathrm{ML}}$, where $N_{a}$ is the surface concentration of adparticles and $N_{\mathrm{ML}}$ is their concentration in monolayer, one has to include adparticles interactions in overlayer. The most important is the dipole–dipole repulsion, which can be taken into account by the replacement of $\varepsilon_{a}$ to $\varepsilon_{a}\left(\Theta \right)=\varepsilon_{a}-{\zeta \Theta }^{3/2}Z_{a}\left(\Theta \right)$, where the dipole-interaction constant $\zeta =2e^{2}l^{2}N_{\mathrm{ML}}^{3/2}\bar{A}$, *e* is the elementary charge, *l* is the adparticle bond length, and $\bar{A}\sim 10$ [21]. It is worthy to note that all the interactions of adsorbed particles lead to the decrease of $\left|Z_{a}\left(\Theta \right)\right|$. The role of the finite temperature effect on $Z_{a}$ is discussed in [22].

The effect of adsorption on the substrate appears in mainly two effects. One is the change Δφ in the work function due to the charge transfer between an adparticle and the substrate. As a result of this transfer, the adparticle acquires charge $Z_{a}$, which may favor $\left(Z_{a}>0\right)$ or prevent $\left(Z_{a}<0\right)$ the escape of an electron from the substrate, thus, lowering (Δφ<0) or raising (Δφ>0) the work function. In the former case, the electron passes from donor adparticle to the substrate; in the latter, it leaves the substrate for acceptor adparticle. The second effect due to adsorption is the change in the surface conductivity of the substrate ΔG. The reason for this effect is twofold. First, the surface carriers’ concentration $n_{s}$ changes as follows: The donor (acceptor) adparticles increase (decrease) the conductivity of the *n*-type substrate or, conversely, decrease (increase) the conductivity of a *p*-type substrate. Second, the adsorbed particles (adparticles) serve as additional scattering centers, which generally must influence the surface mobility $\mu_{s}$ of the carriers. The systematic studies of the simultaneous changes in the surface conductivity and work function began with experimental works on gas molecules adsorption on metal oxides [23,24]. A theory that relates quantities ΔG and Δφ was developed in [25,26], where the following equation was obtained:(4)$$\left|\frac{\Delta G\left(\Theta \right)}{\Delta \varphi \left(\Theta \right)}\right|=\frac{\mu_{s}}{4\pi \mathrm{el}}\equiv \eta$$

Analysis [25,26] of experimental data [23,24] have shown that ratio η does not explicitly depend on the coverage Θ. Thus, we have arrived at the equation $\Delta G={e\mu }_{s}{\Delta n}_{s}+{\mathrm{en}}_{s}{\Delta \mu }_{s}\approx {e\mu }_{s}{\Delta n}_{s}$. We suppose that this equation is of great importance for the resistive-type gas sensors. It is important to underline that the quantities ΔG and Δφ have to be measured simultaneously. 

In [27], we have applied Equation (3) to the analysis of experimental data on gas molecules adsorption on carbon nanostructures (see corresponding references in [14]). This analysis demonstrates a number of inconsistences of published experimental results with Equation (3). Some additional theoretical estimates are given in Appendix A.

## 3. Graphene Film Production Technology

Interest in graphene flared up after the publication of K.S. Novoselova, A.K. Geim et al., In which they demonstrated the possibility of obtaining graphene sheets using micromechanical cleavage of bulk crystalline graphite [28]. This was primarily due to the unique physical and mechanical properties of graphene, such as high thermal and electrical conductivity, high mobility of charge carriers, high Young’s modulus, combination of optical transparency with good electrical conductivity, etc. The listed properties are very attractive from the point of view of possible applications of the material as a basis for nanoelectronics devices [29,30].

Until now, graphene samples obtained by this technology have the best structural perfection. It was on these samples that the most important results were obtained, which confirmed the two-dimensional nature of this material. However, their small size, irregular, and unpredictable geometrical shape in advance do not allow their use in industry. Graphene films obtained by thermal destruction of the surface of silicon carbide are the second in terms of structural perfection. Thus, it is possible to obtain structures up to industrially important dimensions; in this case, the dimensions are limited only by the initial SiC substrate, i.e., up to 6 inches (150 mm). The orientation of the obtained graphene films is specified by a SiC substrate, which makes it possible to obtain graphene films with a large area with a preferential azimuthal orientation of domains. This method is based on the dissociative evaporation (sublimation) of SiC components from the surface of a single crystal and the formation of a graphene film from residual carbon atoms [31] The phenomenon of the formation of a carbon film on the SiC surface has been theoretically analyzed for a long time in [32]. The main advantages of the method for growing graphene on SiC can be attributed to the absence of the need for subsequent transfer of the grown film to an insulating substrate, since growth can be performed on high-resistance SiC substrates with a resistivity of >10^8^ Ω cm. Schottky diodes based on the graphene/SiC structure, obtained by thermal destruction of SiC substrates, are characterized by high uniformity of characteristics.

Graphene films grown by CVD on a metal foil are in third place in terms of their structural perfection [33]. This method makes it possible to obtain graphene of large dimensions, with a quality slightly lower than that obtained by thermal destruction. The disadvantage of the CVD method is the need for subsequent transfer of the material to a dielectric substrate. This procedure increases the chain of technological operations for creating the final device and can also negatively affect the structural perfection of the transferred graphene.

The main disadvantage of the method of thermal destruction of the SiC surface is the high cost of SiC substrates. However, the high structural perfection of graphene obtained by the thermal destruction method, as well as the possibility of growing graphene on high-resistance substrates, neutralize this disadvantage.

Dissociative evaporation or sublimation of SiC is one of the most important processes that determine the growth of crystals and epitaxial films of this material from its own vapors. The work of S.K. Lilov [34] presents a detailed thermodynamic analysis of equilibrium processes in the gas phase during sublimation of SiC in the temperature range of 1500–3150 K. This leads to the fact that the dissociative evaporation of SiC is accompanied by the formation of free carbon in the condensed phase. This carbon accumulates on the surface of the sublimating crystal (substrate or batch grain), covering it with a continuous layer [33]. The process of graphitization of the SiC surface is a fairly common phenomenon that often manifests itself in the growth of bulk SiC crystals by the PVT method. Therefore, to develop the design of a technological unit for the production of graphene on a SiC surface by the sublimation method, the general concept of an equipment designed for the growth of SiC crystals by this method was used. For the specific task of growing carbon films with a thickness of up to 1 nm, the design of the technological unit was developed, which allows precise control of the main technological parameters. The setup diagram is shown in Figure 1. Figure 2 shows a photograph of the setup for the growth of graphene on the SiC surface.

An important step in obtaining a graphene film is the selection of a SiC substrate that will be used for the thermal destruction process.

The first criterion for choosing substrates is their crystal perfection. To obtain homogeneous graphene, substrates with a homogeneous crystal structure are required; therefore, it is preferable to use single-crystal SiC substrates. Various structural defects, such as dislocations, micropores, stacking faults, polytypic inclusions, low-angle boundaries, etc., can also negatively affect the uniformity of graphene growth on the substrate surface; therefore, it is necessary to choose substrates with a minimum density of structural defects.

The second criterion for choosing substrates is the surface roughness on which graphene growth is planned. Today there are several stages of surface treatment of SiC wafers: Grinding, mechanical polishing, and chemical-mechanical polishing.

The third criterion for choosing substrates is the crystallographic orientation of the substrate surface, as well as the angle of deflection of the substrate surface from the basal plane (the so-called misorientation angle α). In our experiments, we used 6*H* and 4*H*-SiC substrates with a minimum misorientation angle (α ~ 0), the growth was carried out on the (0001) face (Si face). We used semi-insulating (high-resistivity substrates), since they did not require the transfer of a graphene film onto a non-conducting substrate in the further manufacture of sensors.

Modern commercial SiC wafers are in standard sizes of 2, 3, 4, or 6 inches. The use of such large samples for laboratory research is not profitable, because, firstly, the cost of the material is quite high, and secondly, the characterization of large samples takes a lot of time. Therefore, in order to carry out experiments on developing the technology of graphene growth, the wafers were cut using special equipment for cutting semiconductor substrates into typical specimens 5 × 5 mm^2^ and 11 × 11 mm^2^ in size. 

The growth of graphene on the SiC surface is accompanied by the sublimation of the Si_x_C_y_ components from the substrate surface. Various surface contaminants or surface irregularities of the substrate can contribute to the sublimation process, leading to nonuniform sublimation of molecules from the substrate surface. For the successful development of graphene growth technology, a necessary condition is a high-quality preparation of the SiC surface, which reduces the effect of contamination and surface inhomogeneities on the sublimation process. 

Pre-growth etching in a hydrogen atmosphere was used for preliminary cleaning of the SiC substrate surface. The method has been known for a long time [8,9] and is used in various technological and epitaxial processes associated with silicon carbide. The essence of the technology lies in high-temperature heating of the SiC substrate in a hydrogen atmosphere. At high temperatures, free carbon formed on the SiC surface binds with hydrogen to form volatile chemical compounds.

To enable the etching of SiC substrates “in situ” before graphene growth, this technology was adapted to a technological unit for the growth of graphene on SiC. The adaptation consisted in changing the composition of the gas mixture in which the etching is performed, as well as in the selection of a certain technological mode of etching.

Typically, for etching, the chamber is purged with either pure hydrogen [35] or with a mixture of hydrogen with other gases, for example, C_3_H_8_ [36]. In our case, we used a gas mixture containing argon (volume fraction 95%) and hydrogen (volume fraction 5%). As known, argon is an inert gas, so it does not take part in chemical processes that occur on the surface of the substrate when it is heated. The choice of a gas mixture with a low percentage of hydrogen content is primarily due to safety considerations when conducting experiments using flammable and explosive gases.

## 4. Study of the Parameters of the Obtained Epitaxial Films

The grown graphene samples were characterized by using a combination of three methods: Raman spectroscopy, atomic-force microscopy (AFM), and Kelvin-probe force microscopy (KPFM).

Raman spectroscopy is widely used as a graphene characterization tool. Analysis of the data obtained by these methods allows one to determine such properties of graphene as number of layers, degree of defectiveness, strain and doping levels, and many more. Figure 3 shows an array of Raman spectra obtained by mapping of a sample area of 12 × 12 μm^2^ with spatial resolution of 1 μm. The measurements were carried out at room temperature using a T64000 Raman spectrometer (Horiba Jobin-Yvon, Lille, France) equipped with a confocal microscope. The spectra were excited using the 532 nm Nd:YAG laser. (Laser Quantum, Stockport, UK). To avoid heating and damage to the sample, the laser power was limited to 4 mW. Two main features in the presented spectra, namely the sharp and intense *G* and 2*D* lines, are originating from the graphene film [37], while the complex background with several maxima in the 1300–1600 cm^−1^ range corresponds to the buffer layer, which is an interface carbon layer located between the graphene film and the 4*H*-SiC substrate [38].

The *D* line, the appearance of which is indicative of presence of structural defects [39], in our case is indistinguishable from the background originating from the buffer layer. This fact indicates the high quality of the graphene film under study.

Double-resonance nature of the 2*D* line allows one to easily distinguish monolayer graphene from bi- and multilayer by analyzing the shape of this line. Figure 3b shows the map of the 2*D* line FWHM distribution. In the regions with FWHM (2*D*) between 35 and 40 cm^−1^, this line had a symmetric Lorentzian form, which pointed to the monolayer nature of graphene in corresponding areas [37]. In the remaining 10% of the map, where the 2*D* line FWHM was increased, the line had an asymmetric shape similar to that reported for bilayer graphene [10] and could be fitted with four Lorentzians.

An AFM topography map of the 12 × 12 μm^2^ area of the sample is shown on Figure 4a. The surface topography and the surface potential distribution were examined with an NtegraAURA (NT-MDT, Moscow, Russia) scanning probe microscope under atmospheric conditions, with NSG11 (NT-MDT, Moscow, Russia) semi-contact probes having a conducting Pt coating. The measurements were performed using a standard two-pass technique in which the surface topography was recorded in the first pass. In the second pass, the surface potential was recorded with amplitude modulation and a probe-surface distance maintained at 20 nm. The surface of the sample is composed of elongated steps with height varying from 1 to 2 nm. The root-mean square (RMS) parameter, which characterizes the surface roughness, is 0.41 nm.

Distribution of surface potential is also capable of providing information on graphene thickness. It is known that the surface potential of bilayer graphene is ~100 mV higher than that of monolayer [40]. The KPFM map demonstrating the surface potential distribution in a 12 × 12 μm^2^ area of the studied sample is shown in Figure 4b. According to the difference of the surface potential values between dark and bright areas, the dark areas should be attributed to monolayer graphene, and the bright areas, to the bilayer one. Bilayer graphene, as follows from the KPFM data, covers approximately 10% of the sample, which is in agreement with the results derived from Raman spectroscopy data.

## 5. Development and Testing of a Graphene-Based Gas Sensor

It was shown in [1] that graphene is capable of sensing the adsorption of even one molecule. As you know, the resistance of a conductor is determined by both the concentration of charge carriers and their mobility. The adsorbed gas molecules, depending on their charge, behave like donors or acceptors. Those change the concentration of mobile charge carriers. In addition, adsorbates create additional scattering centers and change the carrier mobility. As a result, depending on the type of adsorbed molecule, a decrease or increase in the resistance of the film is observed.

To clean the graphene detector, a current of about 10 mA is passed through it this is enough to heat the structure so that the gas particles are desorbed. This cleaning mechanism does not affect the degree of efficiency of gas detection: The process of sorption-desorption of gases is completely reversible, that is, it is a reusable detector. It should be noted that, in [1], for the manufacture of the sensor, graphene obtained by the exfoliated method was used. The graphene films obtained by this technology have the best structural perfection, however they have small sizes and irregular shapes, which makes this technology unpromising for industrial production. The sensor structure was formed on a graphene film using laser photolithography [41,42] (Figure 5). Excess graphene was removed from the substrate surface by etching in an oxygen-argon plasma. Ohmic contacts Ti/Au (5/50) nm were prepared by explosive photolithography after deposition of metals on the photoresist surface by electron beam evaporation. The sensor chip was fixed to the holder together with two resistors. One of the resistors was used to measure the temperature, and the other was used as a heater.

Purified air was used as a carrier gas. Sensor sensitivity (r) was expressed as a percentage, %, and is defined as the relative change in the sample resistance in the presence of a recorded gas in the gas mixture:г = (R − R_o_)/R_o_(5)
where R is the sensor resistance when gas is supplied; R_o_ is the initial resistance in the absence of a detectable gas in the incoming air flow.

Figure 6 shows the relative changes in the resistance of a graphene-based sensor in the presence of NO_2_ in the gas mixture (periods of gas supply are indicated as light gray stripes) at 20 °C. Since the NO_2_ desorption rate at room temperature is very low, annealing at 110 °C was used to return the sensor to its original state after each exposure period [43,44].

For NO_2_ concentration of 10 ppb, the amplitude of the sensor response was about 3% when exposed to the gas mixture for 1 h Such sensitivity of the sensors is quite sufficient for environmental monitoring.

It should be noted that one of the serious disadvantages of the graphene gas sensor is the lack of selectivity. Indeed, it is impossible to tell from the change in conductivity which molecule was adsorbed on the graphene surface. Moreover, some molecules give contributions of the opposite sign, so the total change in resistance can be close to zero.

## 6. Graphene-Based Biosensor: Detection of Influenza Viruses

### 6.1. Concept of Graphene/SiC Biosensors

A distinctive feature of the graphene film is that the molecules or a group of atoms adsorbed on its surface act as a donor or an acceptor, leading to a change in its electronic state (resistance), which can be detected [45]. This specificity of the graphene film, in principle, can be used to create biosensors for the registration (diagnosis) of extremely low concentrations of biomolecules associated with various socially significant diseases at an early stage (hepatitis, oncology, HIV or hemolysis, viral diseases (influenza, coronavirus)).

However, the graphene film itself is not a selectively sensitive sensor and can attach various substances and biomolecules to its surface. To use graphene as a biosensor, a special treatment is used, which increases the selectivity of chemical reactions on the graphene surface and creates additional covalent bonds for chemical reactions with other molecules that need to be detected.

The antibody–antigen (Ab-Ag) interaction is fundamental to the functioning of the human immune system [46]. Reactions Ab-Ag is the reaction of specific binding of the antigen with its corresponding antibodies, leading to the formation of an immune complex. The interaction of Ab-Ag is carried out according to the principle (key-lock) of the three-dimensional spatial complementarity of the outer electron clouds of the antibody and the antigen molecules. In vitro, these reactions are the basis of many immunological methods and are widely used in laboratory practice.

The concept of the graphene/SiC biosensor developed in his work is based on the creation of conditions for a controlled Ab-Ag interaction on the graphene surface in graphene/SiC chips. It is this interaction that leads to a change in the electronic state of graphene (its resistance), which can be registered. The developed concept of a biosensor is universal for the detection of protein compounds and viruses Only complementary (related) antibodies and antigens take part in the interaction reaction, which achieves the selectivity of the biosensor. To provide biosensing ability, graphene/SiC films are undergoing multistep processing and treatments.

### 6.2. Graphene/SiC Sensor Preparation 

Graphene films were grown by thermal decomposition of semi-insulating substrates (0001) ± 0.25° 4*H*-SiC a size of 11 × 11 mm^2^. The growth process was carried out in a graphite crucible with induction heating in an argon atmosphere (720–750 Torr) at a temperature of 1700–1800 °С [47]. The method allows one to obtain high-quality graphene films on a high-resistance substrate of arbitrarily large areas, which is important to process graphene chips (dies) for sensing applications.

The presence of the graphene monolayer on the SiC substrate was confirmed by Raman spectra. A specific feature of the morphology of SiC surface after graphene growth is the presence of terraces on the surface, the width and height of which depends on the conditions of graphene growth. As example, Figure 7 shows three-dimensonal (3*D*) (AFM) images of graphene/SiC samples. A set of elongated terraces having different width are clearly seen in the morphology image of the surface. These steps can facilitate attachment to graphene of large (several nanometers) antibodies of viruses.

After the graphene film growth and characterization, graphene/SiC chips (dies) were processed for manufacturing biosensors. The processing includes standard photolithography using dry etching, metallization, cutting the substrate into individual chips, and mounting them on a holder. A rectangular chip topology was formed on the surface of a graphene/SiC sample by laser photolithography in combination with ion-reactive etching in argon and etching in oxygen plasma. Ti/Au contacts (5 nm/50 nm) were created by vacuum deposition and explosive photolithography. Then, the graphene/SiC sample was cut into individual 1.5 × 2 mm^2^ chips, which after that were mounted on a holder and welded with gold wires. After that, all current-carrying parts of the holder and contact pads of the chip were covered by a protective varnish. The protective varnish is a compound based on thermo reactive resins with various fillers and additives (VT-25-200). Sensing area of the chip was 1 × 1 mm^2^ or so.

The current-voltage (*I-V*) characteristics of the manufactured chips were linear, which indicated ohmic contacts and the absence of potential barriers on the contacts that could affect the measurement results.

For biosensor applications, additional processing or functionalization of the graphene surface in the chip takes place to make it susceptible to interaction with biological molecules. There are several approaches to the modification of chemical reactions on the surface of graphene, one of them is associated with the formation of a covalent bond between graphene and aromatic hydrocarbon groups [48]. This approach, which is called the covalent functionalization of graphene (covalent method), we used in this work. It is chemically reliable and provides ample opportunities for influencing the electronic properties of a graphene film.

In this project, the process of functionalization of the graphene surface in the chip will be carried out in two stages by creating covalent bonds during the deposition of nitrophenyl groups (nitrobenzene, C_6_H_4_NO_2_) and their subsequent reduction to phenylamine groups (aminobenzene C_6_H_4_NH_2_) using a two-steps cyclic voltammetry (CV) process [48].

Figure 8 shows the cyclic voltammograms of the functionalization process. To deposit nitrobenzene at the first step of CV process, a graphene chip mounted on a holder was placed in an anhydrous electrolyte based on a mixture of 4-nitrophenyldiazonium tetrafluoroborate (4NDT) in acetonitrile (C_2_H_3_N) (0.025 mg in 50 mL of acetonitrile) and tetrafluoroborate (tetrabutylammonium) tetrafluoroborate (TBA 1.65 g in 50 mL of acetonitrile). The electrochemical reaction between the graphene surface in the chip and the anhydrous electrolyte was carried out in a cell, of which the design allows purging the electrolyte and the space above it with a clean and dry inert gas (argon) to remove traces of moisture from the ambient air. The process was carried out according to a three-electrode scheme. The graphene surface in the chip was a working electrode, a platinum plate served as a counter electrode, and a silver wire (Ag/Ag^+^) served as a reference electrode.

The reaction of attaching of nitrobenzene occurred during cycling of the potential at the working electrode (the graphene surface) in the range from 0 mV to −600 mV and vice versa. In this case, the process current was recorded. Figure 8a shows typical cyclic voltammograms of the addition (binding) process of the nitrophenyl groups. A decrease in the current with each subsequent cycle indicated the completion of the reaction on the surface of the working electrode. Usually, three cycles were used, until the current was reduced by several times. Then the samples were rinsed in acetonitrile and dried with pure argon.

At the second step of CV process, the reduction of nitrobenzene attached to graphene to aminobenzene was carried out already in an aqueous solution of 0.1 M KCl and ethyl alcohol (9:1) in an open flask also according to a three-electrode scheme. The electrodes were: A standard silver chloride reference electrode (Ag/AgCl) Esr-10101 (“Measuring equipment”, Moscow, Russia), the graphene surface in the chip was a working electrode (anode), and a platinum plate counter electrode. The reduction reaction was also carried out in the mode of cycling the potential on the working electrode from 0mV to −1000 mV and vice versa. After the process, the samples were washed in deionized water and dried in argon. Typical CVs of reduction processes are shown in Figure 7b. A decrease in the current with each subsequent cycle indicated the completion of the reaction on the surface of the working electrode. After functionalization of graphene, the chips were stored in a desiccator in an argon atmosphere.

After functionalization, the deposition (or immobilization) of antibody molecules took place on the created covalent bonds. Then, the graphene surface was passivated (blocked) in a solution of 0.1% BSA in PBS (bovine serum albumin solution in phosphate-buffered saline) for 1 hour, followed by washing in pure PBS for 5 minutes. After that, the graphene chip is ready to apply as a biosensor.

Figure 9 shows an image of the prepared biosensor and the principle of its operation. The detection principle is to create conditions for the antibody-antigen (AB-AG) reaction on the prepared graphene surface. The AB-AG reaction changes the electronic state of graphene (changes the value of the current flow in the graphene channel between two contacts), which can be recorded by electronic devices. In this work, we used the antigen–antibody interaction on a functionalized graphene surface in graphene/SiC chips to investigate their sensitivity for early diagnosis of socially significant diseases like influenza viruses.

### 6.3. Influenza Viruses Sensing Experiments

First, the response of the chip (current through the chip) was investigated during the reaction of complementary antibodies and antigens of the influenza A virus (antibody A-antigen A) on the prepared graphene surface. We used strains of influenza viruses obtained from the Smorodintsev Research Institute of Influenza, St. Petersburg, Russia: influenza virus A/California/07/09 (H1N1pdm09) and influenza virus B/Brisbane/46/15.

At the beginning, antibodies to influenza A or B virus were immobilized on chips with functionalized graphene surface for 3 h at 37 °C. The monoclonal antibodies to influenza A or B viruses used in the experiment are directed to nucleoprotein (NP), a highly conserved major protein of the influenza virus associated with RNA (ribonucleic acid). Then, the graphene surface was passivated (blocked) in a solution of 0.1% BSA in PBS (bovine serum albumin solution in phosphate-buffered saline) for 1 h, followed by washing in pure PBS for 5 min The graphene chips (sensors) prepared in this way were used in the detection experiments.

Before the sensing experiments, two groups of dilutions were prepared in PBS for influenza A virus and influenza B virus at different concentrations from 1 × 10^−15^ g/mL to 1 × 10^−10^ g/mL (1 fg/mL, 10 fg/mL, 100 fg/mL, 1 pg/mL, 10 pg/mL, and 100 pg/mL for total protein), 6 dilutions and 12 tubes in total. Since it was initially assumed that chips based on graphene monolayers could be very sensitive sensors, the range of selected concentrations of antigens to be detected was chosen below the detection limit of the PCR (polymerase chain reaction) test widely used in enzyme-linked immunosorbent assay for detecting viruses. This should confirm the advantage of graphene sensors in terms of sensing ability. The experiments described below showed that this choice of the ultra-low concentration range of antigens was justified, and the chips showed their sensitivity down to a concentration of 1 × 10^−15^ g/mL (1 fg/mL). Note that the work did not set the task of determining the sensitivity limit of the chips being developed for the detection of influenza viruses.

First, the direct reaction of the complementary antibody and antigen on the surface of graphene in the chip was investigated. The prepared graphene sensor was first incubated with pure PBS (washing solution, blank), then its response (current passing through the chip) was recorded at DC conditions when, depending on the sensor resistance, a sequence of DC voltage of 20 mV, 40 mV, 60 mV, and 80 mV, was applied between chip contact pads for 20–30 s. Then, a graphene sensor was incubated into a tube with the PBS solution of an antigen A (or B) virus for 20–30 s and its response was recorded. The chip was sequentially incubated in the different antigen virus solutions one by one from low to high concentration of the antigen virus. First, the direct reaction of complementary antibodies and antigen on the surface of graphene in a chip was investigated. An example of the timing diagrams of the chip response during its sequential incubation in PBS solutions of influenza A virus (AV) antigens is given in Figure 4. EG313-4(A)-AV means a chip (EG313-4) with an antibody of the influenza A virus (A) that was incubated during the experiment in solutions of the antigen of the influenza A virus (AV) and its response was recorded at various DC voltages on the chip. After the antibody A virus immobilization, no special treatment of the graphene surface was occurred.

Figure 10 shows the chip response (current through the chip) from solutions with antigens A virus of different concentrations. The responses are parallel and near stable during measurements at DC voltage of 40 mV supplied to the chip. The responses from antigen solutions in PBS are higher than responses from pure PBS solution. This means that (1) the reaction of antigen attachment to the graphene surface leads to an increase in the conductivity of the graphene channel between the contacts in the chip and (2) the chip has a potential to detect lower that 1 fg/mL concentration of the influenza A virus in PBS solution.

The data in Figure 10a allow plotting the dependence of the chip response on the antigen concentration in the buffer solutions, which are shown in Figure 10b. There is a general tendency to an increase in the chip response (current through the chip) with an increase in the concentration of influenza virus A antigen over the entire range of investigated concentrations.

The data in Figure 10b are well approximated by a logarithmic function with parameter R^2^ = 0.96. The dependency in Figure 10b was plotted on a semi-logarithmic scale, so the data fit looks like a straight dashed line. Parameter R^2^ becomes smaller for a DC voltage of 80 mV due to the scatter of the data. We think this scatter reflects the specificity of reactions on graphene surfaces with terraces, which, according to AFM scanning, have the width of up to 1100 nm for chips of this series (EG313). The logarithmic dependence was observed earlier for graphene-based biosensor when studying its response on egg albumin solutions in PBS [49]. Note, it was assumed that an excess amount of unsaturated bonds (immobilized antibody A virus) remained on the graphene surface of for the attachment of antigen A virus for every incubation of the chip in antigen A virus solutions up to the highest used concentrations. Results in Figure 10 have similar trends at all voltages applied to the chip (see Figure A1 and Figure A2 in Appendix B). Discussion of the results of subsequent experiments will be carried out for data obtained at a voltage of 60 mV on the chip.

In continuation of these studies, additional passivation of graphene in an albumin solution (0.1% BSA in PBS) was performed after immobilization of the antibody during the preparation of the chips for the next experiments. In addition, the chips had different widths of terraces on the graphene surface. The timing diagrams of the chip’s response (current through the chip) were similar to those shown in Figure 10, so they are not shown in the separate figure. However, these diagrams were used to plot the dependence of the chip response depending on the antigen concentration in PBS solutions shown in Figure 6. The concentration dependences of the chip response (current through the chip) were plotted for two chips with different widths of terraces on the graphene surface.

The designations of the chips in this work at different stages of experiments are presented in the Table A1 in Appendix B. The designation of the EG313-41(A)Al-AV chip in Figure 11 means that it is a chip (EG313-41) with an antibody of the influenza A virus (A) and passivated in albumin solution (Al) that was incubated during the experiment in solutions of the antigen of the influenza A virus (AV) in PBS. The same designation is used for chip EG331-61(A)Al-AV. The chips in these experiments had different widths (and heights) of terraces on the graphene surface, which, as it turned out, influenced the responses of the chips.

EG313-41 chip, which has a terrace on the surface of graphene about 1 µm wide, shows the dependence of the increased response of the chip (current through the chip) with the concentration of antigen A in the solution that is well approximated by logarithmic function with the parameter R2-0.94. In contrast, the chip with wider terraces (up to 4 microns) showed a decrease in chip response with an increase in the concentration of virus A virus antigen with a clear saturation of the response at concentrations of 1 × 10^−12^ g/mL and above. In contrast, EG331-61 chip with wider terraces (up to 4 microns) showed a decrease in chip response with an increase in the concentration of virus A virus antigen with a clear saturation of the response at concentration of 1 × 10^−12^ g/mL (1 pg/mL) and above.

During the functionalization of graphene, unsaturated covalent bonds are created on its surface that promote to the attachment (immobilization) of the necessary antibodies. At the detection process, only complementary (related) antigens can react with immobilized antibodies noticeably changing the electronic state of graphene. This is the main principle of operation the sensors that are being developed. The interaction of the remaining non-complementary molecules with the graphene surface does not have the character of a chemical reaction, but rather occurs due to weak van der Waals forces, possibly due to defects and imperfections of the graphene/SiC substrate composition. This interaction can be further screened to reduce its effect on the electronic properties of graphene by passivating the graphene surface with neutral molecules like albumin. However, the attachment of molecules to the graphene surface that are non-complementary to the immobilized antibody can have a significant effect on the resulting chip response (current through the chip).

We believe that a decrease and saturation of the chip response (current through the chip) in Figure 11b indicates an insufficient number of covalent bonds with immobilized antibodies on the graphene surface in the EG331-61 chip with very wide terraces (up to 4 µm).

Too small of a number of sites with an immobilized virus A antibody creates conditions when the virus A antigens, the concentration of which is higher than the A virus antigens, are forced to directly attach to the graphene surface due to van der Waals forces, thus significantly affecting the nature of the change in the chip response. Apparently, the graphene surface treatment and graphene functionalization conditions should take into account the specifics of graphene morphology, in particular, the width and height of terraces on its surface.

Special experiments were performed to demonstrate specificity of the antibody–antigen interaction on the graphene surface. The experiments were carried out in two stages.

The procedure for the first stage of the experiment was the same as described above with one exception. The only difference was that the prepared chip with the immobilized antibody was immersed (incubated) in solutions of non-complementary antigens of viruses. In other words, the chip with the immobilized antibodies of the virus A was immersed in solutions with the antigens of the virus B and its response (current through the chip) was recorded. Similarly, the chip with the immobilized antibodies of virus B was incubated in solutions with the antigen of the virus A.

At the second stage of the experiment, the same chip was immersed for 20 s in other solutions but with virus B antigens (second incubation) that were complementary (related) to the antibody virus B immobilized on the chip. The chip response (current through the chip) was also recorded. After the experiments, the responses of graphene chip (current through the chip) were compared and studied.

As before, the timing diagrams of the chip response (current through the chip) obtained at both stages of the experiment were similar to those shown in Figure 10, which made it possible to plot the dependence of the chip response on the antigen concentration in PBS solutions.

Figure 12 shows the concentration dependences obtained for three chips at the first and second stage of experiments. The chips differed in the width of the terraces on the graphene surface. Similar to already noted earlier, the designation of the EG260-9(B)Al-AV means that it is a chip (EG260-9) with an antibody of the influenza B virus (B) and passivated in albumin solution (Al) that was incubated in solutions of the antigen of the influenza B virus (BV) in PBS during the first stage of the experiment. The designation of the EG260-9(B)Al-AV-BV means that it is a chip (EG260-9) after the first stage of the experiment (EG260-9(B)Al-AV) that was incubated in solutions of the antigen of the influenza B virus (BV) in PBS during the second stage of the experiment.

Figure 12a–c shows the data for the first stage of the experiment, when the antibody immobilized on the graphene surface in the chip was non-complementary to the antigen in the PBS solution. Since the reaction of non-complementary antibodies and antigen should not occur due to the fundamental principles of these reactions [50,51], then, as expected, there is no obvious relationship in these figures, but there is a scatter of the response values over the entire range of the studied antigen concentrations. In this case, the spread between the maximum and minimum response values for the EG260-9 chip with narrow terraces (400–600 nm) reaches less than 200 nA, and reaches 530–550 nm for other chips with a large terrace width.

In this case, the spread between the maximum and minimum response values for the EG260-9 chip with narrow terraces (400–600 nm) reaches less than 200 nA, and up to 530–550 nm for other chips with large terraces width.

The results of the second stage of the experiments are shown in Figure 12d–f. In this experiment, the interaction of complementary antibodies on the graphene surface and antigen in PBS solutions took place. For all chips, the magnitude of the response increased with the concentration of antigens and showed a clear dependence, which was well approximated by the logarithmic function with the parameter R^2^ 0.9–0.97.

For the EG260-9 chip results in Figure 12d, a saturation of the response was observed at the highest studied antigen concentration of 1 × 10^−10^ g/mL (100 pg/mL), which is possibly due to the saturation of antibodies on the graphene surface by the reaction with antigens in a PBS solution. When approximating the experimental results, this point was not taken into account.

For the EG313-15 chip with moderately wide steps on the surface (1 μm), a similar logarithmic dependence was observed in Figure 12f over the entire range of investigated concentrations of virus B antigen in PBS solutions from 1 fg/mL to 100 pg/mL.

The scatter of the data reduced the parameter R^2^ to 0.9 when approximating the experimental results. It should be noted that in this study the EG313 series chips most often showed the dependence of the response on the concentration of antibodies A or B in the form of a logarithmic function (parameter R^2^ more than 0.9) over the entire range of investigated concentrations.

The EG332-13 chip in Figure 12f with the widest terraces on the graphene surface (2 μm) in this experiment showed a markedly increased response (current through the chip) measured at the lowest antigen concentration of 1 × 10^−15^ g/mL (1 fg/mL) in PBS solution. A similar maximum response value was observed for the EG331-61 chip in Figure 11b with terraces up to 4 µm wide when incubated in an antigen solution with the lowest concentration.

We attribute this feature to the fact that in highly dilute solutions the antigen interacts not only with immobilized complementary antibody molecules on the surface, but also directly deposited on the graphene surface if the sites with antibodies are located at a great distance from each other or there are simply few of them. The presented results for chips with wide steps in Figure 11b and Figure 12b prove this assumption.

Note also that the passivation of the graphene surface in chips with albumin is necessary rather to saturate the covalent bonds that have not attached the antigen than to passivate the graphene surface [52]. Real passivation is required with neutral protein molecules to reduce the influence of foreign molecules on the chip response and to maximize the sensor response.

In conclusion, a series of experiments on the detection of virus antigens A and B with chips (sensors) based on graphene monolayer films on SiC was carried out. The sensitivity of the chips for the detection of diluted solutions of virus antigens in PBS solutions in the concentration range from 1 × 10^−15^ g/mL to 1 × 10^−10^ g/mL is shown.

The specificity of the reaсtion antibody A—antigen B and antibody B—antigen A on the graphene surface was demonstrated. The results confirm the principle of operation of the sensor based on the reaction of interaction of a complementary antibody and an antigen on the graphene surface. The surface morphology of a graphene monolayer, in particular the width of characteristic terraces, affects the response of the chip (current through the chip) when detecting influenza viruses.

## 7. Conclusions

The advent of graphene opened a new page in solid state physics—the production and study of two-dimensional materials. The development of research rather quickly raised the question of the possibility of practical application of these materials. It was obvious that the method of mechanical exfoliation, which produced the first samples of graphene, was not suitable for industrial production.

In this work, we considered the most promising technology for producing graphene films, namely, thermal destruction of the silicon carbide surface. This technology makes it possible to obtain films with a sufficiently high structural perfection, and when using semi-insulating SiC, there is no need to transfer the grown graphene to a dielectric substrate. To create devices, one can use post-growth technologies common for semiconductor production.

The presence of electrophysical parameters of graphene such as the maximum ratio of area to volume, a low concentration of carriers in combination with their high mobility, and a low noise intensity makes it possible to obtain supersensitive resistive sensors on its basis. The practical application of graphene gas sensors is hindered by the lack of selectivity in the registration of various gases. The use of an antigen—antibody pair allows one to solve the problem of biosensor selectivity and opens up very wide possibilities for the use of graphene-based sensors in medicine and biology. This approach may lead to the creation of portable biosensors capable of detecting diagnostically significant markers of diseases in biological fluids in the express analysis mode.

## Figures and Tables

**Figure 1 materials-14-00590-f001:**
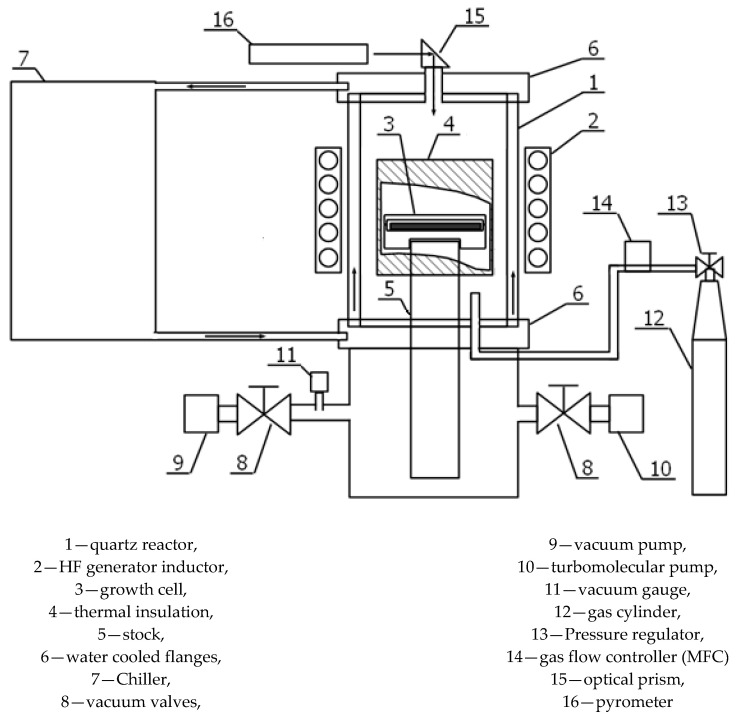
Schematic of a technological setup for the growth of graphene on a SiC surface.

**Figure 2 materials-14-00590-f002:**
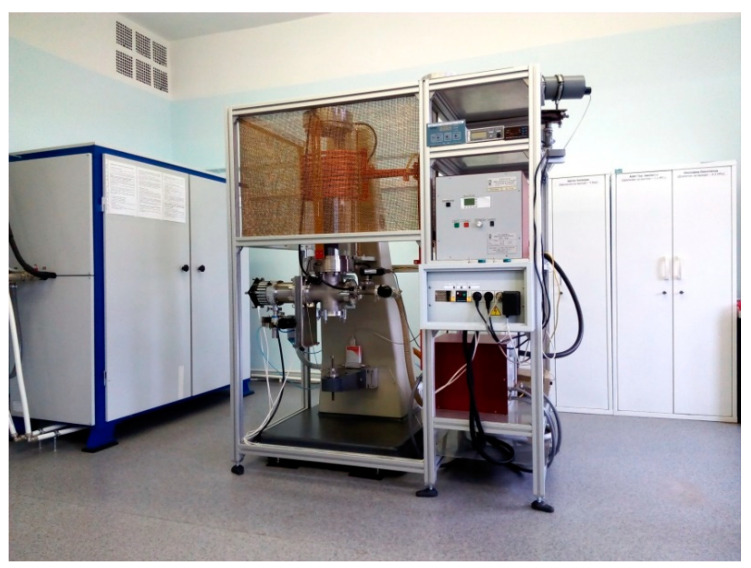
Technological setup for the growth of graphene on the SiC surface.

**Figure 3 materials-14-00590-f003:**
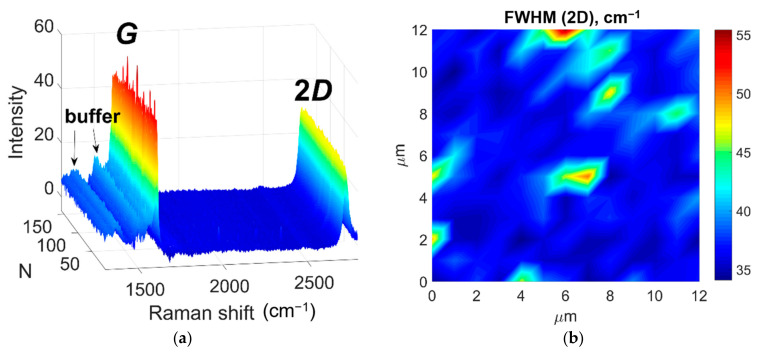
(**a**) Array of Raman spectra obtained from an area of 12 × 12 μm^2^. The spectra are presented after subtraction of the 4*H*-SiC substrate contribution. N denotes the serial number of the spectrum measured during the mapping process. (**b**) Raman map of 2*D* line full width at half maximum (FWHM) distribution.

**Figure 4 materials-14-00590-f004:**
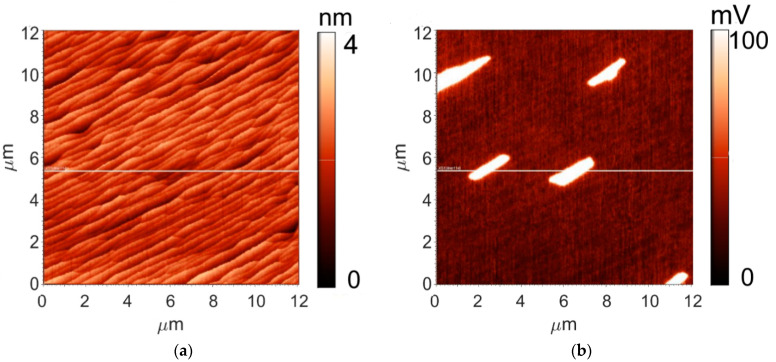
Surface topography (**a**) and surface potential distribution (**b**) maps of the sample under study measured in the same area.

**Figure 5 materials-14-00590-f005:**
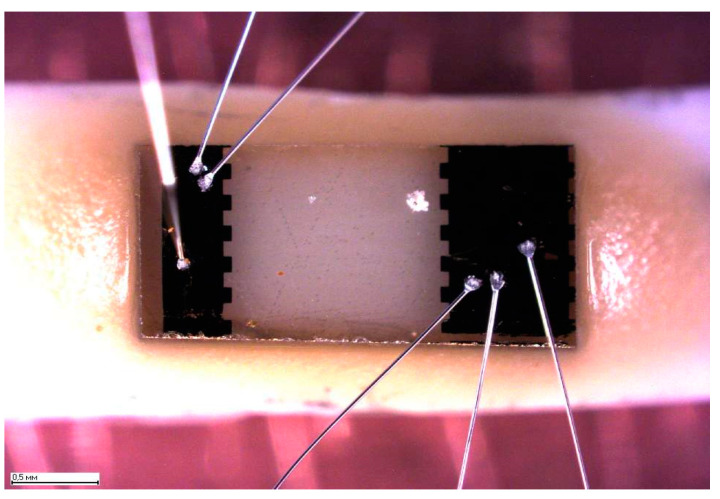
Gas sensor chip. The light area is graphene, the dark area is the ohmic metal contacts with attached wires.

**Figure 6 materials-14-00590-f006:**
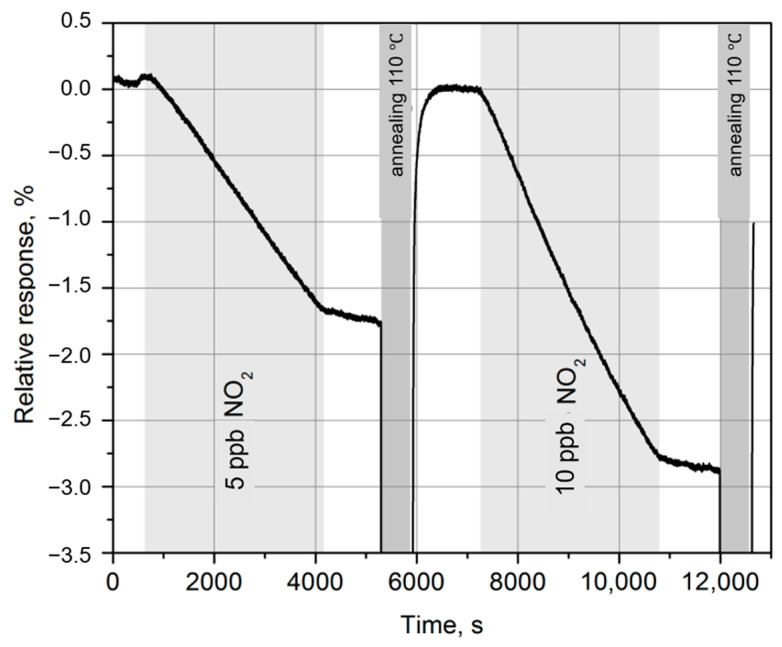
Dependence of the response of a graphene-based gas sensor on the concentration of NO_2_ in the gas mixture at a temperature of 20 °C. Gas supply periods are indicated as light gray stripes, annealing periods are dark gray.

**Figure 7 materials-14-00590-f007:**
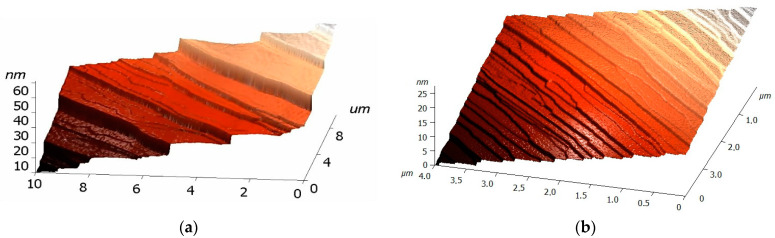
3D atomic-force microscopy (AFM) images of two graphene/SiC samples having different terraces on the surface. (**a**) Terrace width: ~2 um; terrace height: 8–10 nm; (**b**) Terrace width: ~0.4 um; terrace height: 1–3 nm; graphene was grown on SiC substrates at various temperatures and pressures.

**Figure 8 materials-14-00590-f008:**
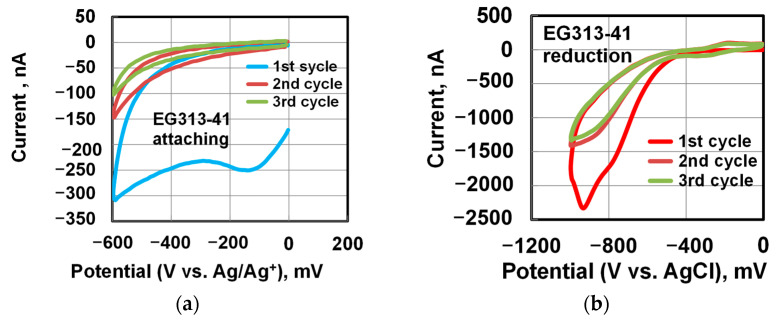
Cyclic voltammograms of the functionalization of the graphene surface in a chip. (**a**) The process of attaching of nitrophenyl groups (a silver wire was the reference electrode), (**b**) the process of reduction of nitrophenyl groups to phenylamine groups (the reference electrode was a standard silver chloride Ag/AgCl electrode, Esr-10101). Three scan cycles were used.

**Figure 9 materials-14-00590-f009:**
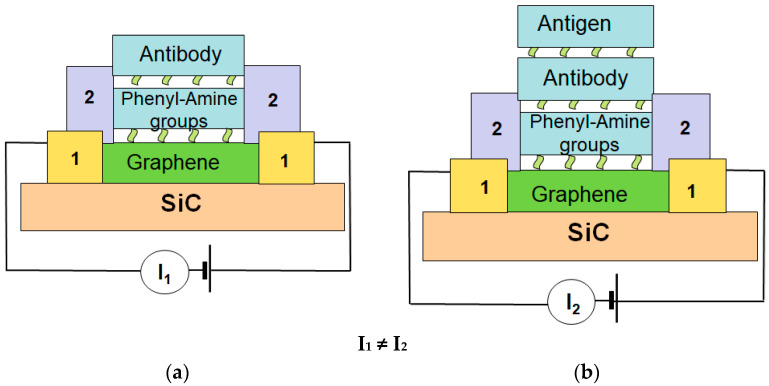
A scheme of the biosensor operation. 1—contact pads, 2—protective layer. (**a**) Sensor with functionalized graphene surface and immobilized antibody on it. (**b**) Detection process. The antigen to be detected chemically attach to the antibodies on the functionalized graphene surface that results in a change of the resistance of the graphene channel, which can be promptly detected by the current passing through the graphene/SiC die. The reaction will not occur if antibody does not match the antigen. It creates selectivity of the biosensor.

**Figure 10 materials-14-00590-f010:**
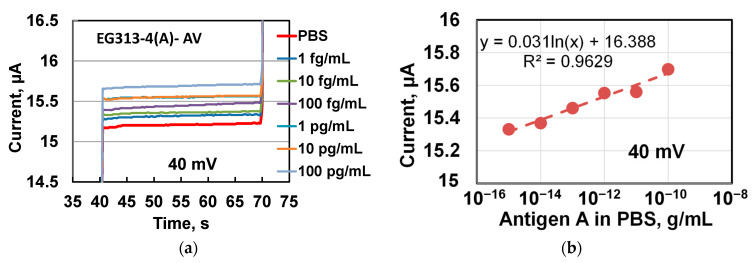
The response of the EG313-4 chip (current through the chip) when it is dipped (incubated) in PBS solutions with the virus A (AV) antigen concentrations (**a**) timing diagrams of the current through the chip at 40 mV on the chip. (**b**) concentration dependence of the chip response. The data is taken from the timing diagram. The dashed line shows the approximation of the data by a logarithmic dependence.

**Figure 11 materials-14-00590-f011:**
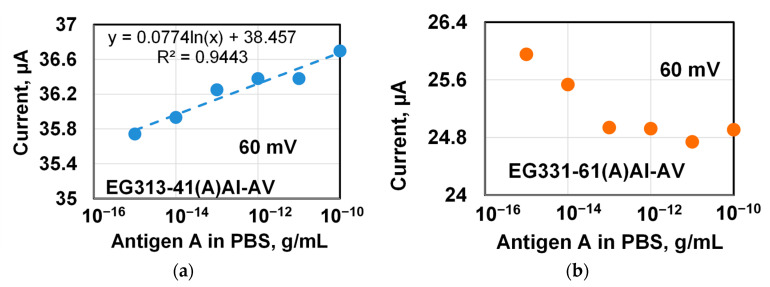
The response of two chips (current through the chip) passivated in bovine serum albumin (BSA) depending on the concentration of the virus A antigen in the PBS solution a DC voltage 60 mV. The width of the terraces reaches 1 µm and 4 µm for chips in (**a**) and (**b**), respectively. The dashed line shows the approximation of the data by a logarithmic dependence.

**Figure 12 materials-14-00590-f012:**
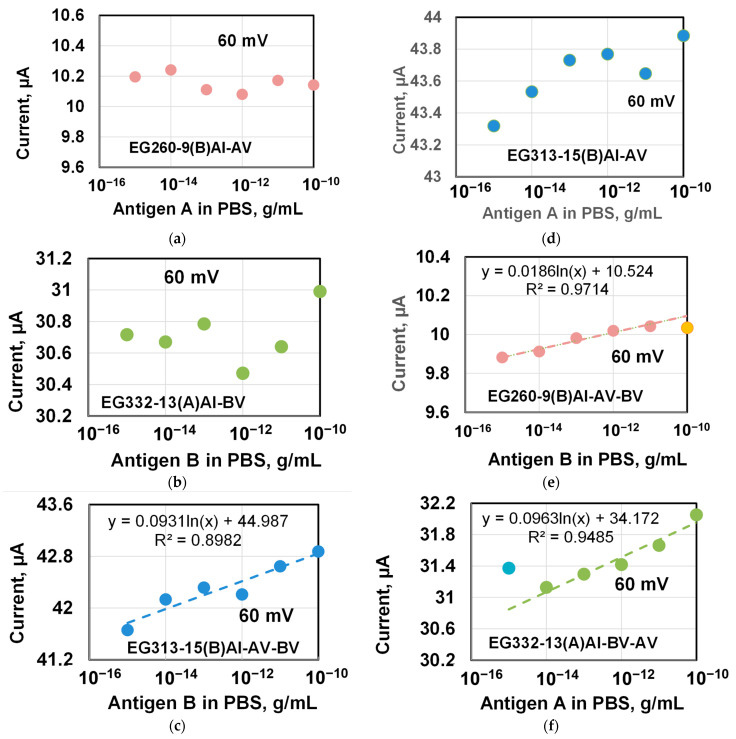
The response of chips (current through the chip) depending on the concentration of influenza A or B antigens in a PBS solution in experiments to demonstrate the specificity of antibody–antigen reactions on the graphene surface. (**a**–**c**) Current through the chip at the first stage of the experiment. The reaction of non-complimentary antibodies and antigen on the surface of graphene in the chip. (**d**–**f**) Current through the chip at the second stage of the experiment. Reaction of complementary antibodies and antigen on the surface of graphene in a chip DC voltage of 60 mV was applied to the chips. The dashed line shows the approximation of the data by a logarithmic dependence. The width of the terraces reaches 0.6 µm, 1 µm, and 2µm for chips EG260-9, EG313-15, and EG332-13, respectively.

## Data Availability

Data is contained within the article.

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
