# Peer review of "Graphene on SiC Substrate as Biosensor: Theoretical Background, Preparation, and Characterization"

_materials, 2021, doi:10.3390/ma14030590_

Round 1

Reviewer 1 Report

Please add following references about graphene sensors grown on insulators:

https://www.nature.com/articles/srep19822

https://pubs.rsc.org/en/content/articlelanding/2018/nr/c7nr06330j/unauth#!divAbstract

In Figure 3, what does 'N' represent? Repeat measurements? Also, what is the buffer? You mention that SiC background was subtracted from Raman signal. 

What are parameters (power and wavelength) used for Raman spectroscopy?

Same for KPFM. What is probe diameter, material, and sample distance?

A majority of DFT derivation and equations could be moved to supplementary section or appendix. Perhaps include a plot of analytical or numerical solution to summarize major findings/predictions.

Content beginning on line 265 is most relevant to sensor operation and provides excellent theoretical background on sensing mechanism. 

Raman spectroscopy should be performed on device after fabrication steps. Does photolithography damage graphene?

Line 334 (NO2 sensor) - was graphene resistance change as a function of temperature studied? Cooling may also reduce graphene resistance, and gas infusion may lead to cooling.  

Line 403 - please give detail about 'protective varnish'

Figure 8 - use different colors to indicate cycle numbers. This will help others to reproduce your method. 

Reduce the number of figures for the viral detection section. I suggest making 1 or 2 informative figures describing the parameters which worked best. Please include the others (e.g., different voltages) in a supplementary section. 

Section 7 should either be greatly reduced (choose 1 or 2 important equations) moved to supplementary.

Conclusion should describe disadvantages of SiC graphene production method and compare pros and cons of other more scalable approaches. For example, cost, equipment, and technical skill required are much greater than chemical exfoliation (graphite to GO to rGO) methods. Also, liquid phase graphene oxide is much easier to coat substrates and allows for deposition after photolithography (e.g., spray, spin coat, or dip through hard mask). Of course, the quality of rGO is much lower than SiC graphene. 

Reviewer 2 Report

In their extensive article entitled “Graphene on SiC substrate as biosensor: theoretical back-ground, preparation and characterization”, Davydov et al. extensively present their merit on manufacturing a chip-scale, graphene-based biosensor. The fabrication process of the sensor (or actually sensors) is in depth described in the manuscript, ranging from the initial graphene production on a SiC substrate, to the chip preparation and the Antibody-Antigen technique used by the Authors to selectively detect different type of influenza viruses (A and B types). Furthermore, the performed test/measurements by the Authors provide sufficient evidence that their sensor is indeed of good quality, potentially having industrial and/or medical applications. I just have a few comments/request for the Authors to include in their revised version of the manuscript, which I, in principle, suggest to be published in Materials.

a. Generally, an in-depth proof-reading of the article is in order since I found some grammatical errors, a lot of typos or even other logical errors (e.g., wrong citation of figures in the main text).

b. I consider the abstract being quite large. The Authors should reduce it significantly, even halving its original extend!

c. Although not an expert in the field, I do know that CVD-grown graphene is extensively used in applications. There is not any mention of this fabrication process in the text (e.g., in section 2). I believe mentioning the process and a few of its advantages/disadvantages is in order, especially when it is put in contrast with the epitaxial graphene growth on SiC used by the Authors.

d. Regarding section 4, I did not understand its significance. I did not find anywhere in the rest of the manuscript a use of the presented equations or the mathematical framework in general. Did the authors use this theoretical framework to, e.g., estimate that their sensor would indeed supersede other commercial techniques before the manufacturing stage? If yes, such important steps should be clearly noted in the text. Generally, I believe this section should be better tied with the rest of the manuscript or else they should remove it completely, given also the fact that it is hard to follow for the reader as well.

e. I was really confused with the sensor names and in general with terminology regarding the antibodies-antigens pairs used in the measurements. The Authors should think of a better way to refer to each sensor and the respective measurements they performed on it as well. Furthermore, the Authors mention that they fabricated a number of sensors but only a few are included in the respective schematics. If this is the case and other measurements were performed but did not fit in the manuscript, I suggest the Authors to include them in a reader-friendly format, e.g., in a table or something similar.

Author Response

Please see the attacchment

Reviewer 3 Report

The work reported in this manuscript is interesting and well presented. So I suggest Accept in present form.
